# Analysis of Oral Health among ADHD-Affected and Non-ADHD Children in Gran Canaria

**DOI:** 10.3390/healthcare12070779

**Published:** 2024-04-03

**Authors:** Roshan Melwani-Sadhwani, Eva Ruth Alonso-Agustín, Audrey Sagols-Ruiz, Ana Isabel Contreras-Madrid

**Affiliations:** 1Department of Dental Sciences/Faculty of Dentistry, University Fernando Pessoa Canarias, 35450 Santa María de Guía, Spain; acontreras@ufpcanarias.es; 2Ministry of Education of the Canary Islands, 35001 Las Palmas de Gran Canaria, Spain; ealoagu@gobiernodecanarias.org; 3Primary Care Management of the Area of Gran Canaria, 35006 Las Palmas de Gran Canaria, Spain; asagrui@gobiernodecanarias.org

**Keywords:** ADHD, oral health status, attention deficit, hyperactivity

## Abstract

A study in Gran Canaria assessed the oral health of 53 children diagnosed with attention deficit hyperactivity disorder (ADHD) and 106 without ADHD, aged 6 to 16 years. Clinical data on dental caries, conservative treatments, and periodontal health were recorded. The main objective of this study was to investigate the prevalence of dental caries in children with ADHD and how their habits and behaviors contribute to their oral health status. The results showed that children with ADHD had a significant increase in dental caries, conservative treatments, and gingival bleeding. A connection with diet was found, though it did not have a negative impact. No increase in periodontal index was observed, and no significant relationship with diet, family factors, or breastfeeding was found. The evidence suggests that children with ADHD should undergo frequent dental check-ups to monitor oral health and receive education on the importance of the quality and frequency of tooth brushing. These findings highlight the need to address oral health in children with ADHD and the necessity for adequate dental care in this group.

## 1. Introduction

Oral health is a critical aspect of public health that is particularly important during childhood and adolescence, when habits affecting long-term oral health are developed. The World Health Organization emphasizes the integration of oral health within the broader context of pediatric health [1].

Attention deficit hyperactivity disorder (ADHD) not only affects behavior and learning but also has significant implications for oral health. Fisher-Owens et al. [2] showed in their study the relationship between ADHD and a high incidence of dental caries and periodontal diseases in children. This relationship can be attributed to various factors, including neglect of oral hygiene, less effective tooth brushing [3,4,5], and a higher propensity for consuming sugars and carbohydrates [6,7] Moreover, recent studies conducted in Spain, such as those by Caleza-Jimenez et al. and Mota-Veloso et al. [8,9], have provided a valuable comparative context that enriches our understanding of the interaction between ADHD and oral health. These insights lay the groundwork for a regional comparison that highlights the necessity of contextualized research within diverse Spanish populations.

ADHD is characterized by a lack of concentration and inappropriate levels of hyperactivity and impulsivity [9,10,11] and is one of the most common childhood conditions, typically diagnosed at school age [12]. This condition significantly impacts family stress, academic performance, and self-esteem [13]. The global prevalence of ADHD is estimated between 3% and 10% in Spanish children and adolescents [14].

This study aims to respond to the findings of experts like Cerrillo-Urbina et al. [15], arguing for the importance of better understanding the relationship between ADHD and oral health in specific contexts to develop effective interventions and public health policies. In exploring the oral health of children with ADHD in Gran Canaria, this study aims to shed light on an aspect that has not been investigated to obtain new results. The importance of this research is reflected in the findings of academics from Maden and İpek [16], who pointed out that children with neurological development conditions such as ADHD face an increased risk of poor oral health outcomes. In addition to this, other studies suggest that specific preventive and educational strategies can significantly improve these outcomes [17,18,19].

### 1.1. Prevalence and Risk of Dental Caries in Children with ADHD

Numerous studies have examined the complex relationship between attention deficit hyperactivity disorder (ADHD) and oral health in children. Maden and İpek [16] found a higher incidence of dental caries in children diagnosed with ADHD. These findings suggest that distinctive ADHD symptoms, such as inattention, hyperactivity, and impulsivity, may contribute to poor oral health.

Children with ADHD face unique challenges in maintaining oral hygiene. Additionally, impulsive behaviors may lead to unhealthy dietary choices, like excessive sugar consumption, increasing the risk of caries. Velázquez-Cayón et al. [20,21] noted a high prevalence of dental caries in this population, reflecting these behaviors.

Children with ADHD are more prone to dental anxiety and behavioral problems, which can complicate their dental visits [22,23,24,25].

The impact of ADHD medication such as psycho-stimulants on oral health is also crucial. According to [16], these medications can have side effects like dry mouth, increasing the risk of dental caries by reducing saliva flow, which is essential for acid neutralization and enamel remineralization.

These findings point to the need for specialized dental approaches for children with ADHD. Dental health professionals should be equipped to handle not only clinical needs but also the behavioral and emotional challenges associated with ADHD. This involves tailored communication strategies, creating a reassuring dental environment, and close collaboration with other health professionals involved in ADHD treatment.

### 1.2. Oral Health and Management of Dental Anxiety in Children with ADHD

ADHD not only affects cognitive abilities and overall behavior but also has a significant impact on children’s oral health and dental experiences. Dental anxiety and behavioral problems present specific challenges for this population, demanding a specialized approach to their dental care.

Maden et al. and Chandra et al. [16,22] shed light on the high prevalence of dental anxiety and behavioral issues in children with ADHD. These challenges may stem from the general anxiety associated with the disorder and difficulties in understanding and cooperating with dental procedures. Dental anxiety can make dental visits extremely stressful, sometimes leading to the avoidance of necessary dental treatments. Additionally, behavioral issues can interfere with the effective execution of dental procedures, compromising the quality and success of the treatment.

Managing ADHD, given its complex nature, requires close collaboration between dentists, pediatricians, psychologists, and other mental health professionals. 

Children with ADHD face unique challenges in dental care, primarily because of dental anxiety and behavioral problems. Implementing behavior management strategies and adapted communication techniques, along with collaboration between various health professionals, can significantly improve these children’s dental experiences, enhancing their quality of life. By effectively addressing these challenges, more efficient and less traumatic dental care can be achieved for children with ADHD.

### 1.3. Quality of Life and Oral Health in Children with ADHD

Oral health-related quality of life (OHRQoL) is a crucial indicator of overall well-being that is especially significant in children with ADHD. Recent research has explored how ADHD affects oral health-related quality of life (OHRQoL) in children.

One study by Cerrillo-Urbina et al. [15] provided substantial evidence of the influence of ADHD on oral health and, consequently, on oral quality of life. This research suggests that children with ADHD experience a less favorable oral quality of life compared with their non-ADHD peers.

Factors contributing to this impact include a higher prevalence of dental caries, challenges in maintaining proper oral hygiene, and side effects of ADHD medications negatively affecting oral health.

These findings underscore the importance of integrating OHRQoL into the treatment plan for children with ADHD. It is vital to adopt a multidisciplinary approach that addresses both oral health needs and behavioral and emotional concerns.

Focusing effectively on the unique dental care challenges faced by children with ADHD can lead to a substantial improvement in their OHRQoL. This requires a comprehensive, collaborative approach that attends to dental needs while considering the psychological and emotional aspects associated with ADHD and, when relevant, ODD.

### 1.4. Family Cohesion, Parafunctional Oral Habits, and Oral Health Literacy in Children with ADHD

Effective oral health management in children with ADHD goes beyond conventional dental treatment, involving key contextual factors such as family cohesion, parafunctional oral habits, and oral health education. These elements play a crucial role in preventing and managing dental issues like caries.

Family cohesion and support are vital to promoting effective oral hygiene in children with ADHD. A stable family environment and effective communication can be pivotal in adopting healthy oral hygiene habits and ensuring regular dental visits. 

Children with ADHD are prone to developing parafunctional oral habits such as bruxism, nail biting, or prolonged pacifier use, which can negatively impact their oral health, leading to issues like dental wear and malocclusion. Early identification and management are crucial to preventing future dental complications.

Education and literacy in oral health are also key. Understanding oral hygiene and the impact of ADHD on oral health is fundamental. Educating parents and caregivers on optimal oral hygiene practices and the potential effects of ADHD medications is vital to mitigate the risks of caries and other dental problems.

Oral health in children with ADHD is a multifaceted issue that benefits greatly from a comprehensive approach. Considering family cohesion, managing parafunctional oral habits, and providing a solid education in oral health are essential steps in improving not only the oral health of children with ADHD but also their overall quality of life.

By assessing the oral health status of children with ADHD, we seek not only to shed light on a less-studied issue but also to highlight the need for tailored preventive and educational interventions. This approach aligns with the recommendations of experts such as [16,17,18,19,20], who have stressed the importance of adapting public health policies and intervention strategies to the needs of children with neurodevelopmental disorders. By establishing a methodological foundation that is rigorous and focused on the specific needs of children with ADHD in Gran Canaria, this study not only contributes to the existing academic body but also lays the groundwork for future intervention programs aiming to improve the quality of life and oral health of this group.

## 2. Materials and Methods

This cross-sectional study was conducted following the Strengthening the Reporting of Observational Studies in Epidemiology (STROBE) guidelines, as established by [26].

### 2.1. Sample

The study was conducted at the Santa Brígida Health Center in Gran Canaria, Spain, serving a child and adolescent population of 1438, from February 2015 to December 2022. Participants included both genders, aged 6 to 16 years, with and without an ADHD diagnosis. Those with other psychiatric conditions associated with ADHD were excluded. To ensure statistical robustness, we calculated the required sample size based on an expected effect size, a power of 0.80, and an alpha level of 0.05, resulting in a minimum of 53 children with ADHD and 106 controls being necessary for this study 

The ADHD diagnosis was confirmed through diagnostic reports provided by a specialized pediatrician. To avoid diagnostic bias, non-ADHD participants were evaluated to confirm their non-ADHD status using the Du Paul Questionnaire for parents/teachers and the Conners ADHD Scales for parents. Out of the initial sample of 58 children with ADHD, 5 were excluded due to the presence of other associated psychiatric conditions.

To characterize the sample (age, sex, parents’ educational level, and absence of a parent) and evaluate oral health habits, the WHO’s oral health questionnaire for children [1] was administered. Additional relevant sociodemographic variables were collected, such as breastfeeding and adherence to the Mediterranean diet, using the KIDMED test [24]. Breastfeeding and adherence to the Mediterranean diet were considered due to their potential influence on general health and development, which could indirectly affect oral health practices and outcomes in children, including those with ADHD.

The WHO’s oral health questionnaire for children [1] includes indicators such as the index of decayed, missing due to caries, and filled teeth in the permanent dentition (DMFT); the index of decay, fillings, and missing teeth due to caries considering surfaces (DMFS); the index of decay, fillings, and missing teeth by surfaces, considering only the first permanent molar (DMFM); the index of decay, fillings, and missing teeth by surfaces, considering only the first permanent molar (DMFMS); the restoration Index (RI); the gingival index (GI); and the periodontal index (PI).

To ensure accuracy in measuring oral health indicators, two researchers initially assessed the indicators in a group of 15 randomly selected participants. Inter-examiner agreement was evaluated using the Cohen kappa statistic (k = 0.93), indicating a high degree of agreement according to [27]. The characteristics of the participants are summarized in Table 1.

### 2.2. Statistical Analysis

A comprehensive analysis of the involved variables was carried out. For continuous quantitative variables, the mean and standard deviations were calculated, while the absolute frequency and percentage were determined for the discrete quantitative and categorical variables. The normality of the variables was assessed using the Shapiro–Wilk test.

To explore the relationships between oral health indicators and other collected variables, bivariate analyses were conducted using the Mann–Whitney U, test Student’s *t*-test, the Kruskal–Wallis test, and the chi-square test. Variables that showed a statistically significant relationship with oral health indicators were incorporated into multivariate regression models to adjust and analyze in depth the relationship between ADHD and oral health status. All statistical tests were performed with a 95% confidence level using the Stata 16.1 software (Stata Corp, College Station, TX, USA).

### 2.3. Ethical Considerations

This study was conducted in accordance with the principles of the Declaration of Helsinki and received approval from the Clinical Research Ethics Committee of the Canary Health Service (CEIC-CHUIMI-2014/746, approval date: 8 January 2015). To ensure ethical participation, informed consent was obtained from all subjects involved in the study. All parents or legal guardians of the minors involved received a detailed information sheet about the study. Additionally, at the time of the consultation, the informed consent was read to them, ensuring they fully understood the study and its implications. It was emphasized that they were free to accept or refuse participation in the study without any consequences, and upon understanding the information, they provided their signed consent for the inclusion of the minors in the research.

## 3. Results

The responses of 159 participants (28.30% females) were analyzed, yielding an average age of 10.81 (Sd = 2.74). The sample of participants with ADHD numbered 53, and the sample of participants without ADHD numbered 106.

Table 2 provides a succinct overview of dietary characteristics, dental hygiene practices, and oral health status of children with and without ADHD. It highlights adherence to the Mediterranean diet, frequency of dental visits, daily tooth-brushing occurrences, and whether they were breastfed or not. Additionally, the table includes key clinical indicators such as the dental caries index (DMFT), gingival index (GI), and periodontal index (PI), offering a comparison between groups with and without ADHD in terms of oral health and related habits.

Regarding oral health indicators, the bivariate analysis presented in Table 3 shows that six of the seven indicators analyzed (DMFT, DMFS, DMFM, DMFMS, RI, and GI) had worse scores in the group of children with ADHD.

Table 4, Table 5 and Table 6 show the results of the bivariate analysis between study variables and oral health indicators. Variables that showed a statistically significant relationship were included in the multivariate regression models along with the main independent variable (i.e., ADHD). The regression analysis confirms that children with ADHD have a more unfavorable score in the six oral health indicators mentioned above. It is worth noting that adherence to the Mediterranean diet, included in all multivariate analyses, was shown to be a protective factor in three of the four oral health indicators (DMFT, DMFS, and DMFM). 

Our results are consistent with the findings of Caleza-Jimenez et al. and Mota-Veloso et al. [8,9], reinforcing the evidence of a significant association between ADHD and oral health indicators in children. This concurrence underscores the broader relevance of our findings within the Spanish context and suggests common underlying factors across different regions.

## 4. Discussion

Comparing our findings with similar studies conducted in Spain, such as those by [9,10], we observed consistent patterns that emphasize the relationship between ADHD and oral health issues in children. Although these studies did not specifically focus on the population of Gran Canaria, their conclusions regarding the link between ADHD and oral health align with our results, suggesting a consistent trend within the Spanish context. This alignment of findings with prior studies is significant, as it contributes to a broader understanding of ADHD’s impact on oral health, suggesting consistent patterns across diverse Spanish regions. This deepens our comprehension of how geographical and cultural contexts might influence the manifestation and management of oral health issues in children with ADHD.

This study represents a significant advancement in the understanding of oral health in children and adolescents with ADHD in Spain. The detailed assessment of the DMFT index, DMFS index, DMFM index, DMFMS index, RI, and (especially) GI reveals poorer results in the group of children with ADHD. These findings are consistent with previous studies indicating a higher propensity for dental problems in this population [28,29,30]. However, unlike previous studies, our analysis incorporated a broader range of variables, such as sociodemographic aspects, family cohesion, and dental visits, whose importance has been highlighted in the recent literature [10,11]. This advancement is particularly important, as it addresses a critical gap in the literature, providing specific insights into the Gran Canaria population and enhancing our contextual understanding of the ADHD–oral health relationship across varied geographic and cultural settings.

This study demonstrates that increases in the DMFT index, DMFS index, and RI are related to ADHD and age. While the relationship between ADHD and caries is a well-documented phenomenon [16], the correlation with age suggests the need for continuous dental monitoring and care throughout a child’s growth. This reinforces the idea that ADHD not only affects behavior and learning but also fundamental aspects of overall health and well-being, such as oral health. Additionally, an increase in the DMFM index is associated with the presence of ADHD, a higher number of dental visits, and the absence of a parent. These findings partially align with previous studies that did not find family cohesion factors to be associated with oral health [10,11] but differs in the relationship between dental visits and oral health. The interaction between these factors and ADHD indicates a complex model where oral health is influenced not only by neurological conditions but also by socio-economic and familial environments, underscoring the necessity of a holistic approach in their dental care. It is necessary to develop effective personalized strategies based on the findings, such as adaptations in dental communication techniques, anxiety management strategies, and integrated approaches involving pediatricians and psychologists and teachers, as a more inclusive and effective response can be provided for this vulnerable population [19].

Regarding periodontal problems, we identified an increase in GI associated with ADHD, age, and a daily brushing frequency of three or more times, which is consistent with previous studies related to increased gingival bleeding in children with ADHD [28,29]. These studies also reported parental aversion to dentists, bruxism, and inadequate brushing practices, which are aspects that were not examined in our study but are relevant to interpreting our results. Although we found an adequate daily brushing frequency, the quality and duration of brushing may not be sufficient. However, no significant differences were found for the PI, a result that aligns with the findings of Maden and İpek [16] and suggests that ADHD does not have a significant impact on periodontal health.

Concerning dietary habits, although previous studies have documented an increase in the consumption of cariogenic foods in children with ADHD [3], our results suggest that the type of diet does not negatively influence the dental variables studied. This aligns with the findings of Mota-Veloso et al. [5], suggesting that the impact of dietary habits on oral health may be less significant than previously thought.

Regarding breastfeeding, although it has been documented that children with ADHD who do not receive adequate breastfeeding are more likely to experience feeding problems [31], we did not find a relationship in our study. This could indicate that factors related to breastfeeding and early feeding are not determinants in the development of oral problems in children with ADHD.

In summary, our findings underscore the importance of an integrated approach to the oral healthcare of children with ADHD. Given the increasing global prevalence of ADHD [32], it is vital to implement prevention and management strategies that address not only oral hygiene but also dietary habits and regular access to dental services. Frequent dental visits and specific education on the quality and timing of brushing are recommended. Furthermore, it is crucial to consider family dynamics and sociodemographic factors in managing the oral health of these children. The findings of our study highlight the need for specific and tailored interventions to improve oral health and, ultimately, the quality of life of children with ADHD [33,34].

Comparing our results with other studies, both nationally and internationally, makes us reconsider the importance of taking into account factors specific to countries and regions, as well as cultural aspects, in oral health research. This will help develop more effective and targeted interventions not only in children with ADHD but with other conditions.

## 5. Conclusions

After a comprehensive analysis of the data and considering the complexity of the factors involved, this study provides relevant conclusions in the field of oral health in children and adolescents with ADHD.

This study shows that children and adolescents with ADHD have significantly higher rates of oral health problems, including cavities and periodontal issues, compared with their peers without ADHD. The DMFT index, DMFS index, DMFM index, DMFMS index, RI, and (especially) GI were higher in the ADHD group.

In addition to ADHD, factors such as age, family cohesion, dental visit frequency, and diet were shown to influence children’s oral health. In particular, age and dental visit frequency emerged as significant factors.

Despite previous concerns about dietary and hygiene habits in children with ADHD, this study did not find a direct correlation between the type of diet and oral health. However, the quality and duration of brushing, although not specifically measured in this study, may be areas of interest for future research.

The findings underscore the need for a personalized approach to dental care for children with ADHD. Regular dental visits and proper education on oral hygiene are crucial to preventing and managing oral health issues in this population.

This study highlights the need for further research to explore the relationship between ADHD and oral health in-depth, considering factors such as brushing quality and the duration of breastfeeding, which did not show a significant relationship in this study.

Given the increasing prevalence of ADHD and its clear association with poorer oral health outcomes, it is imperative to consider oral health a critical component of the overall well-being of children with ADHD.

This study not only confirms the necessity for specialized dental care for children with ADHD, as observed in other Spanish studies, but also highlights the significance of considering regional variability in future research and intervention programs. By acknowledging and addressing these regional differences, we can enhance the efficacy of public health strategies aimed at improving the oral health and overall well-being of children with ADHD.

In conclusion, this study has successfully met its objective by conclusively demonstrating that ADHD significantly influences the prevalence of dental caries and overall oral health in children, underscoring the importance of personalized and preventive approaches in the dental care of this population.

### 5.1. Limitations of the Study

Sample size: Although the study provided valuable insights, the sample size was relatively limited, which could affect the generalization of the results to larger or more diverse populations.

Sample characteristics: The research focused on a specific population in Gran Canaria, Spain, implying that the results may not be directly applicable to different geographical or cultural contexts.

Controlled variables: While several relevant factors were considered, such as family cohesion and dental visit frequency, other aspects like the quality of brushing and the duration of breastfeeding were not directly measured, which could influence the results.

Subjective dietary assessment: The assessment of dietary habits was based on questionnaires, which may be subject to self-reporting biases.

### 5.2. Future Research

Future research should focus on longitudinal studies examining the progression of oral health in children with ADHD over time as well as the evaluation of specific interventions aimed at improving oral hygiene and reducing dental anxiety in this population.

## Figures and Tables

**Table 1 healthcare-12-00779-t001:** Sociodemographic characteristics.

Characteristics	No ADHD *n* = 106	ADHD *n* = 53
Age (Mean, Sd)	10.81 ± 2.75	10.81 ± 2.76
Gender (*n*, %)		
Male	76 (71.70)	38 (71.70)
Female	30 (28.30)	15 (28.30)
Father’s level of education (*n*, %)		
No studies	1 (0.94)	1 (1.89)
Basic studies	54 (50.94)	29 (54.72)
Vocational training	15 (14.15)	10 (18.87)
University studies	35 (33.02)	12 (22.64)
Mother’s level of education (*n*, %)		
No studies	0 (0.00)	1 (1.89)
Basic studies	38 (35.85)	19 (35.85)
Vocational training	23 (21.70)	18 (33.96)
University studies	45 (42.45)	15 (28.30)
Absence of either parent (*n*, %)		
No	87 (82.08)	31 (58.49)
Yes	19 (17.92)	22 (41.51)

**Table 2 healthcare-12-00779-t002:** Characteristics related to adherence to the Mediterranean diet and oral health indices of the children and adolescents in the sample.

KIDMED Test (*n*, %)		
Very low-quality diet	6 (5.66)	10 (18.87)
Need to improve the dietary pattern	59 (55.66)	28 (52.83)
Optimal Mediterranean diet	41 (38.68)	15 (28.30)
Number of visits to the dentist (*n*, %)		
Has not gone	27 (25.47)	10 (18.87)
From 1 to 3 times	68 (64.15)	33 (62.26)
4 or more times	11 (10.38)	10 (18.87)
Frequency of daily brushing (*n*, %)		
From 1 to 2 times	81 (76.42)	42 (79.25)
3 or more times	25 (23.58)	11 (20.75)
Breastfeeding (*n*, %)		
No	35 (33.02)	15 (28.30)
Yes	69 (65.09)	38 (71.70)
DMFT (Mean, Sd)	0.39 ± 0.97	1.55 ± 2.07
DMFS (Mean, Sd)	0.55 ± 1.43	2.51 ± 3.93
DMFM (Mean, Sd)	0.34 ± 0.85	1.10 ± 1.47
DMFMS (Mean, Sd)	0.42 ± 1.22	1.66 ± 2.65
RI (Mean, Sd)	0.08 ± 0.26	0.28 ± 0.41
GI (Mean, Sd)	26.57 ± 29.41	46.58 ± 37.19
PI (*n*, %)		
Absence of pathological signs	47 (44.34)	14 (26.42)
Bleeding occurs upon probing	41 (38.68)	30 (56.60)
Presence of calculus	18 (16.98)	9 (16.98)

Index of decayed, missing due to caries, and filled teeth in the permanent dentition (DMFT); index of decay, fillings, and missing teeth due to caries considering surfaces (DMFS); index of decay, fillings, and missing teeth by surfaces, considering only the first permanent molar (DMFM); index of decay, fillings, and missing teeth by surfaces, considering only the first permanent molar (DMFMS); restoration index (RI); gingival index (GI); and periodontal index (PI).

**Table 3 healthcare-12-00779-t003:** Bivariate analysis and multivariate regression of oral health indicators in children with and without ADHD.

	DMFT	RI	DMFM	DMFS	DMFMS	GI	PI
**Part A. Bivariate analysis of the study variables with oral health indicators**							
Age	0.000						
Gender	NS						
Father’s level of education	NS						
Mother’s level of education	NS						
Absence of either parent	0.0077						
KIDMED test	0.0263						
Number of visits to the dentist	NS						
Frequency of daily brushing	NS						
Breastfeeding	NS						
**Part B. Multivariate Regression**							
Coefficient	0.94	0.18	0.48	1.73	0.83	16.80	
Sd	0.25						
T	3.78						
*p*-value	0.000						
Confidence Interval (CI) 95%	0.45–1.44						
**Model Information**							
*N*	155						
R^2^	20.01%						
*p*-value	0.000						

**Table 4 healthcare-12-00779-t004:** Multiple linear regression evaluating the presence of ADHD, age, absence of a parent, and the KIDMED test in the DMFT variable.

Predictor Variable	Coefficient	Sd	t	*p*-Value	CI 95%
**ADHD**	0.94	0.25	3.78	0.000	0.45 to 1.44
**Age**	0.09	0.04	2.11	0.036	0.01 to 1.17
**Absence of either parent**	0.27	0.26	1.04	0.300	−0.25 to 0.79
**KIDMED test** (reference: very low-quality diet)					
Need to improve the dietary pattern	−0.94	0.39	−2.43	0.016	−1.70 to −0.18
Optimal Mediterranean diet	−1.02	0.41	−2.51	0.013	−1.82 to −0.22
*Model Data*: *n* = *155*, *pseudo R*^2^ = *20.01%*, *p-value* = *0.000.*					

**Table 5 healthcare-12-00779-t005:** Multiple linear regression assessing the presence of ADHD, age, KIDMED test, and the absence of either parent in the DMFS variable.

Predictor Variable	Coefficient	Sd	t	*p*-Value	CI 95%
**ADHD**	1.73	0.46	3.89	0.000	0.85 to 2.61
**Age**	0.19	0.07	2.56	0.012	0.04 to 0.34
**KIDMED test** (reference: very low-quality diet)					
Need to improve the dietary pattern	−1.36	0.69	−1.98	0.049	−2.72 to −0.00
Optimal Mediterranean diet	−1.45	0.72	−2.01	0.046	−2.88 to −0.02
**Absence of either parent**	0.06	0.47	0.13	0.898	−0.87 to 0.99
*Model Data*: *n = 155*, *pseudo R*^2^ = *18.28%*, *p-value* = *0.000*.					

**Table 6 healthcare-12-00779-t006:** Multiple linear regression assessing the presence of ADHD, age, gender, KIDMED test, absence of either parent, and the number of visits to the dentist in the DMFM variable.

Predictor Variable	Coefficient	Sd	t	*p*-Value	CI 95%
**ADHD**	0.48	0.19	2.57	0.011	0.11 to 0.85
**Age**	0.02	0.03	0.47	0.640	−0.05 to 0.08
**Gender** (reference: male)					
Female	0.11	0.19	0.56	0.574	−0.27 to 0.48
**KIDMED test** (reference: very poor-quality diet)					
Need to improve the dietary pattern	−0.87	0.29	−2.96	0.004	−1.45 to −0.29
Optimal Mediterranean diet	−0.79	0.32	−2.48	0.014	−1.41 to −0.16
**Absence of either parent**	0.46	0.20	2.33	0.021	0.07 to 0.86
**Number of visits to the dentist**					
From 1 to 3 times	−0.06	0.22	−0.28	0.780	−0.49 to 0.37
4 or more times	0.69	0.30	2.33	0.021	0.11 to 1.28
*Model Data*: *n = 154*, *pseudo R*^2^ = *25.36%*, *p-value* = *0.000*.					

## Data Availability

The data of this study are kept by the corresponding author and can be requested by interested parties to the corresponding author.

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
