# Peer review of "Analysis of Oral Health among ADHD-Affected and Non-ADHD Children in Gran Canaria"

_healthcare, 2024, doi:10.3390/healthcare12070779_

Round 1
Reviewer 1 Report
Comments and Suggestions for Authors This is an interesting epidemiological study investigating the mental condition of Attention Deficit Hyperactivity Disorder (ADHD) in a patient population of 159 children (ages 6-16 years old) in the region of the Gran Canaria, in Canary Islands. Children both with and without ADHD were clinically evaluated form the presence of dental caries, periodontal status or the presence of restorations. Interestingly, the data showed that the presence ADHD correlated significantly with all of the clinical measures of disease or restoration studied. An association with diet was also demonstrated, though it didn´t appear to be of a significant impact. The study concludes that monitoring of the oral health status of children with ADHD would require additional dental check-ups as well as oral health education and increased frequency of tooth brushing in order to maintain a healthy status of their oral cavity. Overall, this is an interesting study highlighting an important societal aspect of oral health in a vulnerable population. The topic is rather understudied, and therefore this work can be considered original. The controls in the patient population are appropriate and there are no identifiable methodological drawbacks. The conclusions are consistent and aligning with the data presented. The recommendations provided are based on the conclusions. One gap to be considered that could add to the subject area is that, since this study is performed in a regional cohort, it should be discussed more extensively in relation to similar cohorts in other regions or countries, should such studies be available in the literature. The references used are appropriate, but as mentioned, more effort should be placed in identifying other papers in the literature that may allow for more regional comparisons.
Author Response
Dear reviewer, thank you for your valuable feedback and suggestions. In an enclosed reply letter you will find the response to your comments.
We hope that we have been able to address the changes and suggestions you have made
Kind regards

Reviewer 2 Report
Comments and Suggestions for Authors
Thank you for the opportunity to consider this interesting work.
The manuscript submitted for potential publication in Healthcare aim to explore the complex relationship between ADHD and the prevalence of dental caries, the unique behavioral and dental anxiety challenges these children face, and how these factors affect their quality of life.
Dear Authors,
You analyse very important aspects of oral health. However, there are a number of improvements which are required before it can be accepted.
1. Abstract
There is no aim of your work. Please, add it.
2. Aim of the work
Your aim is very ambitious, but the longest I have ever read. Please, choose the most important aspect and formulate a concise aim. Otherwise, you seem to describe everything and nothing at the same time.
3. Citation in the paper
„The World Health Organization and many studies, such as those by [1]…” I would suggest writing
„ by Smith et al. „ or „The World Health Organization and many studies, such as those by [1], emphasize the integration of oral health within the broader context of pediatric health. [1]”
4. Line 114 „Recent research has delved into how ADHD and associated disorders like Oppositional Defiant Disorder (ODD) impact OHRQoL in children.”
I would not discuss the other disorder, as your work is about ADHD. Please, remove it .
5. Lines 64-168
This part seems to be good for section discussion or for review. You have original work, so remove this part from the introduction or summarise in a few sentences.
6. What do you mean by „assuming a null hypothesis of no differences between the case and control groups.”?
Your long introduction suggested that there are some differences, you even cite the other works. It has no sense.
7. „The WHO Oral Health Assessment Form for Children was used (2013), which includes indicators such as the index of Decayed, Missing due to caries, and Filled Teeth in the permanent dentition (DMFT); the index of Decay, Fillings, and Missing teeth due to caries considering surfaces (DMFS)…”
Please give citations for this paragraph.
8.
„To characterize the sample (age, sex, parents' educational level, absence of a parent) and evaluate oral health habits, the WHO's oral health questionnaire for children (2013) was administered. Additional relevant sociodemographic variables were collected, such as breastfeeding and adherence to the Mediterranean diet, using the KIDMED test [24].
The authors should explain why breastfeeding seems important for 6-year-old children. What is a relations between the Mediterranean diet and ADHD?
9.Result section
„The sample of participants with ADHD was 53, as 5 children were excluded for presenting other associated psychiatric conditions…” Do not mention that you have excluded the patients in this section. You should do it in the methods section.
„Table 1: Sociodemographic characteristics related to adherence to the Mediterranean diet..”
First thing- Sociodemographic characteristics mean basic data concerning the groups. Why do you suddenly describe the Mediterranean diet? Is it a work concerning that aspect? Place the remaining data in the following tables.
Table 2 is not clear. It is not possible to read data from it.
„It is worth noting that adherence to the Mediterranean diet, included in all multivariate analyses, was shown to be a protective factor in three of the four oral health indicators” Was it the aim of your work?
Tables 3 to 5 are not clear. What information do you want to introduce?
Why do you write in the text about the tables from 1 to 7? There is no such a table as 6 .
10. section Discussion
It is too superficial.
11. Conclusions
This section is too long. There is no aim, so it can not be verified.
Author Response

(The authors gave the same response as above.)

Reviewer 3 Report
Comments and Suggestions for Authors Following extensive review I wish to recommend the above for publication following the corrections embedded in the manuscript. Thank you for the collaboration
Author Response

(The authors gave the same response as above.)

Round 2
Reviewer 2 Report
Comments and Suggestions for Authors
Dear authors,
Thank you for your hard work.
There are a lot mistakes that should be corrected: many repetitions, a lack of linguistic fluency in citing works, and strikethroughs within the text . All of them make the paper difficult to read and understand.
Please, read the work carefully and make some corrections to maintain scientific soundness.
Some examples of modifications that can make your work better:
„Oral Health in Children and Adolescents with ADHD..”- the aim in the title
Abstract: „assessed the oral health of 159 children aged 6 to 16 years, both with and without ADHD”- the aim is ADHD, so the number of examined patients is 53. This number should be in the abstract.
For the analysis, there is a question concerning the number of patients with the disease. Is number 53 sufficient to draw any conclusions? It should be calculated
The World Health Organization and many studies emphasize the integration of oral health within the broader context of pediatric health [1].” Citation to „many studies „ is missing
Please read other papers to learn how to introduce the citation. You modified the sentence with citation no 1, so the same correction must be done throughout the manuscript.
Some examples:
„Researchers like [2]”
„Moreover, recent studies conducted in Spain, such as those by [8-9],”…
„This study aims to respond to experts like [16]”
„ADHD is characterized..” Maybe starting the introduction by definition would be a good idea
Aim
„This study aims to respond to experts like [16]”
„ By assessing the oral health status of children with ADHD in Gran Canaria, this study aims”
„As a synthesis, this study aims to”
Choose one aim of the study
Prevalence and Risk of Dental Caries in Children with ADHD
„Numerous studies have examined the complex relationship” – this paragraph can be a beginning of your work
„As [21] noted a high prevalence of dental caries in this population, reflecting these behaviours” there is no grammar
Repetitions:
Line 97
ADHD not only affects cognitive abilities and overall behavior but also has a significant impact on children's oral health
Line 124
Studies by [16] have provided substantial evidence of the influence of ADHD and ODD on oral health
Author Response
Dear Reviewer:
We would like to express our deepest appreciation for the time and effort you have put into evaluating our manuscript. We greatly appreciate your constructive comments and detailed criticism, which have been instrumental in improving the quality and clarity of our study. We recognize the importance of each point raised and have taken careful steps to address them thoroughly in reviewing our manuscript.
Below we present our responses to each of the comments, detailing the modifications made to the manuscript to reflect their valuable suggestions.
Attached main manuscript version R2
The feedback report to the reviewers
MDPI English Editing Service Certificate

Reviewer 3 Report
Comments and Suggestions for Authors Have looked at both the revised manuscript and the cover letter and am extremely delighted with the response of the authors My only observation is that the study is more aligned to case control design than cross sectional study with ratio of 1:2. Finally, I would like to recommend the manuscript for publication Comments on the Quality of English Language Minor editing of English language requiredAuthor Response
Dear Reviewer:
We would like to express our deepest appreciation for the time and effort you have put into evaluating our manuscript. We greatly appreciate your constructive comments and detailed criticism, which have been instrumental in improving the quality and clarity of our study. We recognize the importance of each point raised and have taken careful steps to address them thoroughly in reviewing our manuscript.
Below we present our responses to each of the comments, detailing the modifications made to the manuscript to reflect their valuable suggestions.
Attached main manuscript version R2
The feedback report to the reviewers
MDPI English Editing Service Certificate
